# Transcriptomic Analyses of the Hypothalamic-Pituitary-Gonadal Axis Identify Candidate Genes Related to Egg Production in Xinjiang Yili Geese

**DOI:** 10.3390/ani10010090

**Published:** 2020-01-06

**Authors:** Yingping Wu, Xiaoyu Zhao, Li Chen, Junhua Wang, Yuqing Duan, Haiying Li, Lizhi Lu

**Affiliations:** 1College of Animal Science, Xinjiang Agricultural University, Urumqi 830000, China; wyp_941208@163.com (Y.W.); zxy_950831@163.com (X.Z.); wangjunhua0608@163.com (J.W.); DYQ150325@163.com (Y.D.); 2Institute of Animal Husbandry and Veterinary Science, Zhejiang Academy of Agricultural Sciences, Hangzhou 310021, China; chenli0429@163.com

**Keywords:** Yili goose, hypothalamic-pituitary-gonadal axis, transcriptome, differentially expressed genes

## Abstract

**Simple Summary:**

Transcriptome is the study of gene expression at the RNA level. In this study, high-throughput transcriptome sequencing technology was used to compare and analyze the gene expression differences in the hypothalamus, pituitary, and ovary tissues of Xinjiang Yili geese with high and low egg production. We preliminarily screened 30 candidate genes related to egg production regulation of Xinjiang Yili geese, including the calcium signaling pathway. Therefore, this study laid a theoretical foundation for further elucidating the molecular mechanism of egg production of Yili geese in Xinjiang.

**Abstract:**

The study was conducted to investigate the transcriptomic differences of the hypothalamic-pituitary-gonadal axis between Xinjiang Yili geese with high and low egg production and to find candidate genes regulating the egg production of Xinjiang Yili geese. The 8 selected Xinjiang Yili Geese with high or low egg production (4 for each group) were 3 years old, with good health, and under the same feeding condition. High-throughput sequencing technology was used to sequence cDNA libraries of the hypothalami, pituitary glands, and ovaries. The sequencing data were compared and analyzed, and the transcripts with significant differences were identified and analyzed with bioinformatics. The study showed that the transcriptome sequencing data of the 24 samples contained a total of 1,176,496,146 valid reads and 176.47 gigabase data. Differential expression analyses identified 135, 56, and 331 genes in the hypothalami, pituitary glands, and ovaries of Xinjiang Yili geese with high and low egg production. Further annotation of these differentially expressed genes in the non-redundant protein sequence database (Nr) revealed that 98, 52, and 309 genes were annotated, respectively. Through the annotations of GO (Gene Ontology) and KEGG (Kyoto Encyclopedia of Genes and Genomes) databases, 30 candidate genes related to the egg production of Xinjiang Yili geese were preliminarily selected. The gap junction, focal adhesion, and ECM-receptor interaction signaling pathways were enriched with the hypothalamic, pituitary, and ovarian differentially expressed genes, and the calcium signaling pathway was enriched with the pituitary and ovarian differentially expressed genes. Thus, these pathways in the hypothalamic-pituitary-gonadal axis may play an important role in regulating egg production of Xinjiang Yili geese. The results provided the transcriptomic information of the hypothalamic-pituitary-gonadal axis of Xinjiang Yili geese and laid the theoretical basis for revealing the molecular mechanisms regulating the egg-laying traits of Xinjiang Yili geese.

## 1. Introduction

The Xinjiang Yili goose is an excellent and unique poultry species in Xinjiang, China. It is the only small and medium-sized domestic goose breed domesticated from the grey goose in China [1]. The Yili goose can fly, tolerates coarse feed, displays disease, cold, and heat resistance, and has good meat quality. The Ministry of Agriculture of China has put the Yili goose on the National Livestock and Poultry Breed Conservation List. The Xinjiang Yili Goose has low egg production [2]. An adult Xinjiang Yili goose only produces an average of 8–12 eggs annually [3]. At the same time, its nesting instinct is very strong, which severely restricts its population expansion.

Transcriptomics is an important part of functional genomics research. Transcriptome sequencing (RNA-seq) technology can detect the overall transcriptional activity of a particular species at a single nucleotide level, thus that all transcript information of the species in a certain state can be obtained comprehensively and quickly [4,5]. RNA-seq technology has been widely used to detect new transcripts [6], identify differential genes, and discover transcript structural variations, such as alternative splicing [7,8] and gene fusions [9,10]. The reproductive endocrine system and reproductive activities of birds are mainly regulated by the hypothalamic-pituitary-gonadal (HPG) axis [11]. On this basis, more and more scholars began to use transcriptomics to study the reproductive performance of geese. Liu [12] showed 19, 110, 289, and 211 genes were differentially expressed during the laying and nesting periods in the hypothalami, pituitary glands, ovaries, and follicles of the Tianfu meat geese, and found oxytocin-neurophysin (*OXT*), chordin-like protein 1 (*CHRDL1*), and growth hormone (GH) were differentially expressed in the pituitary, suggesting that they might be new candidate genes associated with reproductive performance of geese. Gao [13] found 48 transcripts were up-regulated during the pre-laying period, 180 transcripts were up-regulated during the laying period. And the identified genes included serine/threonine-protein kinase (*AMPK*), heat shock protein 70 (*HSP70*), and NADH dehydrogenase 1 (*ND1*). Ding [14] conducted a transcriptome study on the ovaries of Sichuan white geese at the pre-laying and laying periods and found that 5 differentially expressed genes (DEGs) might play a role in the high fecundity of Sichuan white geese. At present, there is no report on the molecular mechanisms that regulate the egg production of Xinjiang Yili geese. In order to improve the egg production performance of Xinjiang Yili geese, high-throughput transcriptome sequencing technology was used to compare and analyze differentially expressed genes of Xinjiang Yili geese with high or low egg production and identified candidate genes that are related to egg production of Xinjiang Yili geese. Our study laid a theoretical foundation for elucidating the molecular mechanisms regulating egg production of Xinjiang Yili geese and improving the genetic breeding of Xinjiang Yili geese.

## 2. Materials and Methods

### 2.1. Sample Collection

According to the continuous and complete egg production records (The egg-laying period is from February to June each year) and the pedigree data, we selected 4 high-egg production (HEP) and 4 low-egg production (LEP) Yili geese with an age of 3 years old, which had similar body weights (3.20 ± 0.30 kg) and were under the same feeding conditions. The HEP Yili geese laid 14, 16, 18, and 19 eggs, respectively, and the LEP Yili geese laid 5, 6, 7, and 5 eggs, respectively.

After the goose was sacrificed, their entire hypothalami, pituitary glands, and ovaries (which comprised the whole ovary including the small and large yellow follicles) were collected immediately. The hypothalamic, pituitary, and ovarian samples of the HEP geese were labeled as X21-24, C21-24, L21-24, respectively; the hypothalamic, pituitary, and ovarian samples of the LEP geese were labeled as X01-04, C01-04, L01-04, respectively. After washing with PBS once or twice, the samples were immediately put into individual tubes containing RNA preservation solution. The samples were labeled and stored at 4 °C overnight and then stored at −80 °C the next day until they were used for the extraction of total RNA. The experimental animals were provided by Hengxin Industrial Co., Ltd. of Ermin County, Xinjiang, China. All of the animals mentioned in this study were approved by the Ethics of Xinjiang Agriculture University (Approval number: 2017011).

### 2.2. Feed and Feeding Management

Feeding experiments were carried out at the Yili Geese Breeding Base in Emin County, Xinjiang. The test geese were conventionally raised in a single circle. The breeding geese houses had egg nests, food troughs, ground sports grounds, and water sports grounds. The experimental geese were fed twice at a rate of 200 g per goose per day (8:00 and 16:00 h each day). Eggs were collected and recorded after feeding.

### 2.3. Total RNA Extraction

The total RNA was extracted using Trizol (Invitrogen, Carlsbad, CA, USA) by following the manufacturer’s protocol, and 4 measures were used to ensure the quality of the samples was good enough for transcriptome sequencing. First, agarose gels were used to analyze the extent of RNA degradation and whether the samples were contaminated. Then, RNA was assessed for quantity and quality using a Nanodrop ND-1000 spectrophotometer (Implen, Weslake Village, USA) and Agilent 2100 Bioanalyzer (Agilent Technologies, Santa Clara, CA, USA) according to the manufacturer’s instructions.

### 2.4. Construction of cDNA Libraries and Illumina Sequencing

After the samples passed the quality control, the cDNA libraries were constructed according to the instruction of the Illumina kit (Illumina, San Diego, CA, USA). Next, we used Qubit 2.0 for the preliminary quantification of the constructed libraries and diluted the libraries to 1 ng/uL, and then used Agilent 2100 to determine the insert sizes of the libraries. If the insert sizes met the expectation, we used qRT-PCR to determine the effective concentrations of the libraries. To ensure the quality of the library, the effective concentration of a library should be higher than 2 nM.

After the libraries were qualified, they were pooled and sequenced using an Illumina HiSeq 2500 platform according to the effective concentrations and the requirement of target data quantity. Paired-end sequencing with the 150 bp sequencing read length was performed. The sequencing was done by Novogene (Beijing, China) Bioinformatics Technology Co.Ltd. 

### 2.5. Sequencing Data Analyses

The original image data files obtained from the Illumina HiSeq 2500 platform were transformed into raw sequenced reads by Consensus Assessment of Sequence and Variation (CASAVA pipeline v2.0, Illumina Inc., San Diego, CA, USA) base calling analyses. In order to ensure the quality of bioinformatic analyses, the raw reads were filtered to obtain clean reads. The steps for data processing were as follows: First, the adaptor sequences in the raw reads were trimmed; second, reads with more than 10% unknown nucleotides or with more than 50% low-quality bases (bases quality ≤20) were removed. The follow-up analyses were based on the obtained clean reads. Sequence alignment software HISAT 2.0.4 (Johns Hopkins University, Baltimore, MD, USA) was used to align the sequences of each sample with the reference genome (GRCg6a-galGal6-Genome-Assembly-NCBI.https://www.ncbi.nlm.nih.gov/assembly/GCF_000002315.6) to obtain the position of the clean reads on the reference genome and the barcode sequence information unique to the sequenced samples.

The gene expression levels of each sample were analyzed by HTSeq (version 0.6.1) software. The Union model was chosen, and FPKM (Fragments Per Kilobase of exon per Million fragments mapped) values were calculated. Next, we used the DESeq R package (version 1.10.1) [15] software to standardize the gene expression of different samples and then calculated the *p* values according to the model and finally corrected the multiple hypothesis tests (FDR). After the correction, for pituitary glands and ovaries tissue, the Padj of 0.05 and log2 (Fold_change) with no limitations served as the threshold of significance for differential expression. Only one candidate gene was screened when Padj <0.05 gene was selected as the differential expression gene in the hypothalamus. Therefore, the adjusted threshold for differentially expressed genes in the hypothalamus was *p-*value < 0.005. The differentially expressed genes were analyzed by GOseq software (Illumina, San Diego, California, USA) [16], and the pathway analyses were done by KOBAS (KO Based Annotation System) [17].

### 2.6. Real-Time PCR Validation of Sequencing Results

Randomly selected from the transcriptome sequencing results, 15 differentially expressed genes related to HEP and LEP of Xinjiang Yili geese were used for fluorescence-based quantitative validation. Oligo 7.0 software (Molecular Biology Insights, CO, USA) was used to design primers (Appendix A). SYBR GREEN reagent (TaKaRa) was used to amplify the target gene and internal reference gene (beta-actin) mRNAs on a ROCHE 480 quantitative PCR instrument (Eppendorf, Hamburg, Germany). The PCR reaction system (20 μL) included the 2 μL total RNA as the template, 1 μL upstream primer (10 μmol/L), 1 μL downstream primer (10 μmol/L), 10 μL 2× master mix, and 6 μL ddH2O. The reaction condition was as follows: 95 °C for 15 min, followed by 40 cycles of 95 °C for 10 s, 58 °C or 60 °C for 20 s, 72 °C for 20 s. Quantitative expression results were calculated according to the cross point (CP) values, and the relative expression levels were calculated according to the 2^−∆∆Ct^ method [18]. Primer sequences are given in Appendix A.

## 3. Results

### 3.1. Evaluation of RNA and Sequencing Data Quality

The Nanodrop (Implen, Weslake Village, USA), Qubit 2.0 (Life Technologies, California, USA), and Agilent 2100 Bioanalyzer (Agilent Technologies, Santa Clara, CA, USA) were used to determine the purity, concentrations, and integrity of 24 RNA samples. Appendix A shows that the RNA quality of the samples was good, and the purity and integrity of the samples met the requirements of sequencing library construction.

The quality assessment of the output data of 24 samples is shown in Appendix A. According to the table, 579,563,136 and 596,933,010 reads were obtained by sequencing Xinjiang Yili geese with HEP and LEP. We obtained a total of 176.47 gigabases of data, with at least 5.98 gigabases for each sample. The Q20 and Q30 of each sample were at least 95.10% and 88.57%, respectively, and the GC contents (calculate the percentage of the total number of bases g and C in the total number of bases) were between 48.28% and 50.57%. These results indicated that the quality of transcriptome sequencing results met the needs of subsequent analyses.

The statistics of the sequencing read alignments of the 24 samples to the reference genome are shown in Table 1. According to the statistic results, 72.51% to 77.98% of the clean reads were mapped to the reference genome, among which 71.08% to 77.22% of the clean reads were uniquely mapped. The results showed that there were enough reads of each sample mapped to the reference genome, and the selected reference genome was suitable.

### 3.2. Identification of Differentially Expressed Genes

Gene expression displays temporal and spatial patterns. Both the external stimuli and internal environment can affect gene expression. Under two different conditions, genes with significant differences at expression levels were called differentially expressed genes (DEGs). Padj < 0.05 was used as the criteria for identifying differentially expressed genes in the pituitary and ovary, and *p-*value < 0.005 was used as the criteria for determining differentially expressed genes in the hypothalamus. In this study, the sequencing data of the HEP and LEP Xinjiang Yili geese were compared. A total of 135 differentially expressed genes were identified in the hypothalamus, including 79 up-regulated genes and 56 down-regulated genes. 56 differentially expressed genes were identified in the pituitary, including 25 up-regulated genes and 31 down-regulated genes. A total of 331 differentially expressed genes were identified in the ovary, including 312 up-regulated genes and 19 down-regulated genes (Figure 1, Figure 2 and Figure 3, Table 2).

### 3.3. GO and KEGG Pathway Enrichment Analyses of Differentially Expressed Genes

Blast2 GO software was used to compare the differentially expressed genes and annotate GO functions. The GO annotation includes the enrichment of molecular functions, biological processes, and cellular components [19] (Figure 4, Figure 5 and Figure 6).

The hypothalamic differentially expressed genes were mainly associated with the biological process terms of organophosphate biosynthesis, single-organism development, and development; the cell component term of BBSome and molybdopterin synthase complexes; the molecular functions of anion transmembrane transporter activity, organic anion transmembrane transporter activity, and nickel cation binding. The pituitary differentially expressed genes were mainly associated with the biological process terms of protein metabolism, protein oligomerization, and anion transport; the cellular component term of Nem1-Spo7 phosphatase complexes; the molecular functions of catalytic activity, nucleotide binding, nucleotide phosphate binding, etc. The ovarian differentially expressed genes were mainly associated with the biological process terms of regulation of cellular process, cellular response to stimulus, cell communication, signal transduction, and single organism signaling; the cell component terms of extracellular matrix, proteinaceous extracellular matrix, etc.; the molecular functions of ion channel activity, substrate-specific channel activity, passive transmembrane transporter activity, and calcium ion binding.

Besides, the GO terms of steroid biosynthetic process, steroid hormone-mediated signaling pathway, reproduction, developmental process, and G-protein coupled receptor signaling pathway were found to be associated with the hypothalamic DEGs; the terms of developmental and G-protein coupled receptor signaling pathway and calcium ion binding were found to be associated with the pituitary DEGs; the terms of developmental process, reproduction, and reproduction process were found to be associated with the ovarian DEGs. The differentially expressed genes on the above terms may be involved in the regulation of the reproductive traits of Xinjiang Yili geese. From the above GO terms, 19 differentially expressed genes were selected, including 14 up-regulated genes and 5 down-regulated genes, which could be used as candidate genes associated with the egg production of Xinjiang Yili geese (Table 3).

The pathway enrichment analysis was based on the KEGG database, and a hypergeometric test was used to identify pathways in which the differentially expressed genes were significantly enriched by comparing them with the reference genes of the whole genome. The enrichment analysis results are shown in Figure 7, Figure 8 and Figure 9. The hypothalamic differentially expressed genes were mainly enriched in terms of metabolic pathways, phagosome, sphingolipid metabolism, tight junction, gap junction, focal adhesion, and ECM-receptor interaction signaling pathway; the pituitary differentially expressed genes were mainly enriched in terms of neuroactive ligand-receptor interaction, sphingolipid metabolism, regulation of actin cytoskeleton, metabolic pathways, gap junction, focal adhesion, ECM-receptor interaction, and calcium signaling pathway, etc. The ovarian differentially expressed genes were mainly enriched in terms of focal adhesion, ECM-receptor interaction, vascular smooth muscle contraction, phagosome, tight junction, gap junction, regulation of actin cytoskeleton, calcium signaling pathway, and MAPK signaling Pathway, etc. The differentially expressed genes of the hypothalamus, pituitary gland, and ovary were all enriched in terms of gap junction, focal adhesion, ECM-receptor interaction signaling pathway, and the differentially expressed genes of both the pituitary and ovary were enriched in the calcium signaling pathway, indicating that these pathways played an important role in regulating egg production of Xinjiang Yili geese. Eleven differentially expressed genes were selected from the associated pathways, all of which were up-regulated genes. These genes are candidate genes related to the egg production of Xinjiang Yili geese (Table 3).

### 3.4. Fluorescence Quantitative Polymerase Chain Reaction

In order to validate the differentially expressed genes identified by the transcriptome sequencing, 15 genes were randomly selected and confirmed by qRT-PCR using beta-actin as the internal reference gene. The results showed that the expression trends of the 15 genes were consistent with those of transcriptome sequencing results. Therefore, the transcriptome sequencing results are reliable and can be further studied and analyzed (Figure 10).

## 4. Discussion

The egg production trait of the goose is a trait with low heritability. It is time-consuming to improve the egg production performance of geese by traditional breeding methods. The application of molecular biology technologies provides a new way to improve the egg production performance of geese. RNA-Seq can reveal the expression, structural characteristics, and regulatory profiles of genes by analyzing the transcriptomes of different cells or organs [4], thus identifying candidate genes for the egg production traits. In this study, a high throughput sequencing technique was used to sequence the transcriptomes of the hypothalamic-pituitary-gonadal axis of Yili geese with high and low egg production. Of the differentially expressed genes, 135, 56, and 331 were identified in the hypothalamus, pituitary, and ovary, respectively. Some of the differentially expressed genes (*DRD1, IGF1, ADCY3, SLC4A4, MRAP*) are involved in signaling pathways related to reproductive performance, such as gap junction, adhesion, ECM-receptor interaction, calcium signaling pathway, MAPK signaling pathway, GnRH signaling pathway, oocyte meiosis signaling pathway, etc.

Dopamine (DA) is a major neurotransmitter in the central nervous system, which plays an important regulatory role in the nervous, endocrine, and reproductive systems, and its biological effects are mediated through its receptors. Dopamine receptors belong to the G protein-coupled receptor family, which consists of 7 transmembrane regions. Five dopamine receptors were found until today: D1, D2, D3, D4, D5. D1 and D5 are D1-like receptors, which induced the increase of intracellular cAMPs after activation. D2, D3, and D4 are D2-like receptors, which decrease the level of intracellular cAMP [20]. Niu Shuling [21] found that subcutaneous injection of cAMP significantly improved the laying rate of Hailan hens, a commercial egg-laying hen. Young Ren [22] showed that dopamine could stimulate the secretion of vasoactive intestinal peptide and prolactin by stimulating *DRD1* to regulate the reproductive performance of birds. Xu [23] reported that there were 7 SNPs in the CDS region of the *DRD1* gene in chickens. The haplotypes of G123A and T198C were significantly correlated with the egg production traits and nesting ability of chickens, and the C1107T and G123A loci were correlated with the hatching rate of chickens. Wang [24] found that the mutation of the *DRD1* gene, C681T, was significantly correlated with the pre-laying weight of ducks, and the mutation of A765T was correlated with egg production traits of ducks. Wang Cui [25] scanned the SNPs of the CDS sequence of the duck DRD1 gene. Five SNPs were detected: T189C, C507T, C681T, A765T, and A1044G. The variation loci were significantly correlated with the reproductive traits, egg production, and fat metabolism of ducks. Our study also indicates that *DRD1* is up-regulated in the ovaries of HEP geese and may be involved in the regulation of the calcium signaling pathway. Neurotransmitters activate DRD1, stimulate the activation of adenylate cyclase, and increase intracellular cAMP levels, thus possibly improving the egg production rate.

Insulin-like growth factor-I (*IGF-I*) is a metabolic hormone secreted by granulosa cells. It can stimulate mitosis of follicular granulosa cells, promote the secretion of oxytocin and progesterone from the ovary, stimulate hormone synthesis and secretion from membrane cells, and Leydig cells of the testis, and synergistically work with FSH and estrogen to promote the synthesis of progesterone and estradiol. *IGF-I* is highly homologous to insulin and is one of the important proteins in the growth axis of animals. It plays an important role in regulating cell growth and differentiation [26]. The binding of *IGF-I* to its receptor can promote the proliferation of granulosa cells, maintain the function of aromatase, increase the synthesis of estrogen, and promote the further development of follicles [27]. The results of follicular culture in vitro showed that *IGF-I* could significantly increase the number and thickness of SYF granular and membrane cells in laying hens [28]. Zhu Wenqi [29] analyzed the effects of *IGF-1* on the reproductive performance of Wenchang chickens. The results showed that *IGF-1* was involved in calcium metabolism in vivo and had a significant effect on the numbers of eggs produced by Wenchang chicken at 300 and 400 days, and it was also related to the continuous egg production traits of Wenchang chickens. Yu Mingyue [30] found that there was a correlation between *IGF-1* gene expression and egg production traits in Shiqiza Chicken. The results of our experiments indicated that *IGF-1* was up-regulated in the ovaries of HEP geese, and its role in focal adhesion, oocyte meiosis, and progesterone-mediated oocyte maturation may regulate the reproductive performance of Xinjiang Yili geese by participating in oocyte capacitation and maturation.

Adenylate cyclase (ADCY) is a key signaling molecule downstream of G protein-coupled receptors, which are widely distributed in mammalian cells. This enzyme catalyzes adenosine triphosphate (ATP) to produce cyclic adenosine phosphate (cAMP) and release pyrophosphate. Among all the ADCY subtypes, 9 subtypes are closely related to *ADCY3* [31]. According to Thomas [32], ADCY is localized on the ovarian cell membrane and is a marker enzyme for the separation of ovarian cell membranes. Lee [33] found this enzyme in the ovary of rats. Animal studies have shown that *ADCY3* plays an important role in the regulation of the hypothalamus of rats [34]. In addition, ADCY3 is highly expressed in olfactory epithelial cells [35] and is primarily responsible for the detection of odors. It was found that the absence of *ADCY3* could lead to the deficiency of odor-induced signals and the weakening of maternal behaviors in mice. In addition, *ADCY3* is also associated with normal sperm cell development, sperm functions, and maintenance of male fertility [36]. The results of our experiments show that the *ADCY3* gene is up-regulated in the ovaries of HEP geese and may participate in a variety of signaling pathways: Calcium signaling pathway, GnRH signaling pathway, oocyte meiosis, progesterone-mediated oocyte maturation, tight junction, gap junction, purine metabolism and so on.

Animal reproduction is an extremely complex physiological process that is constrained by exogenous and endogenous factors [37]. In the female reproductive tract, epithelial cells of the uterus and fallopian tube are capable of secreting high concentrations of HCO^3−^, which is important for sperm capacitation and the subsequent fertilization processes [38]. The SLC4 family are very important HCO3-transmembrane transporters. *SLC4A4,* also known as electrosodium bicarbonate cotransporter isoform 1 (NBCe1), is a bioelectric Na^+^/HCO_3_^−^ cotransporter [39] and is widely expressed in a variety of tissues, including the central nervous system [40], the male and female reproductive tracts (the epididymis, and vas deferens in the male reproductive tract, and the endometrial and fallopian tube epithelium in the female genital tract) [41,42]. Liu [41] detected mRNA expression of 5 NBCe1 (*SLC4A4*) isoforms, NBCe1-A to -E, in the reproductive tract of mice, among which NBCe1-D and NBCe1-E are novel isoforms. Studies by Gholami K [43] have shown that estradiol (E2) can up-regulate the expression of *SLC4A4*, which is translocated on the apical and basolateral membranes of the uterine and glandular epithelium under the influence of E2. The basolateral and apical NBCe1 is involved in mediating E2-induced uterine HCO_3_-secretion. By RT-PCR and other techniques, Wang et al. [44] found that nbce1 was expressed in the female genital tract, where nbce1 may participate in HCO_3_^−^ secretion of endometrial epithelial cells. In this study, the *SLC4A4* gene was up-regulated in both the pituitary glands and ovaries of HEP Yili geese and may be involved in the regulation of the above pathways to affect the egg production performance of Xinjiang Yili geese.

Melanocortin receptor 2 (MC2R) belongs to the class A of 7 transmembrane alpha-helix G protein receptors. It regulates the circadian rhythm of steroid hormone secretion and stress-induced changes through CA/cAMP/PKA signal transduction pathway [45]. MC2R functions are strictly dependent on its accessory proteins (*MRAP*), and the functional expression of MC2R can only be achieved in the presence of *MRAP* [46]. According to the recent research results, there are two types of MRAP: MRAP1 and MRAP2 [47,48]. Rouault [49] demonstrated that *MRAP2* regulates the activity of *Orexin*
*Receptor 1 (OX1R)* and interacts with various GPCRs to control food intake and energy expenditure. The data of our experiment showed that MRAP was down-regulated in the pituitary glands of HEP Yili geese. This is consistent with our observation that the Xinjiang Yili geese with LEP generally weighed heavier. Therefore, *MRAP* may play a role in egg production by regulating the goose appetite.

## 5. Conclusions

The results of our study showed that we obtained 1,176,496,146 valid reads and 176.47 gigabases of data from the transcriptomes of 24 samples after quality control. Differential gene analyses showed that 135 genes were differentially expressed in the hypothalami of the high- and low-egg yields Yili geese, including 79 up-regulated genes and 56 down-regulated genes; and 56 genes were differentially expressed in the pituitary glands of the high- and low-egg yields Yili geese, including 25 up-regulated genes and 31 down-regulated genes. A total of 331 differentially expressed genes were identified in the ovaries of the high- and low-egg yields Yili geese, of which 312 were up-regulated and 19 were down-regulated. According to the annotations of GO and KEGG analyses, 30 candidate genes related to the egg production of Xinjiang Yili geese were preliminarily selected. It was found that the adhesion spot, ECM receptor interaction, gap junction, and Ca^2+^ signaling pathway may play an important role in regulating the egg production of Xinjiang Yili geese by the hypothalamic-pituitary-gonadal axis.

## Figures and Tables

**Figure 1 animals-10-00090-f001:**
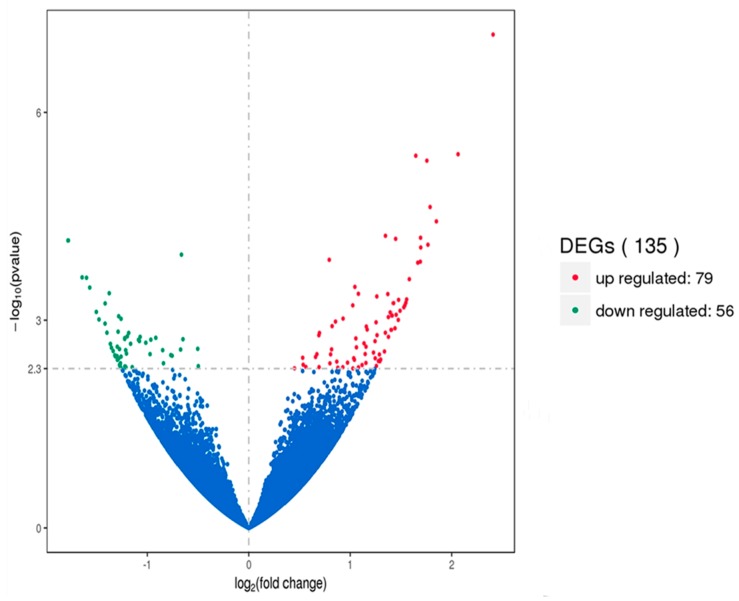
Volcano plot of differentially expressed genes in the hypothalamus. DEGs were filtered using a *p-*value < 0.005 as a threshold. Red spots represent up-regulated genes, and green spots indicate down-regulated genes. Blue spots represent genes that did not show obvious changes between the HEP and LEP samples.

**Figure 2 animals-10-00090-f002:**
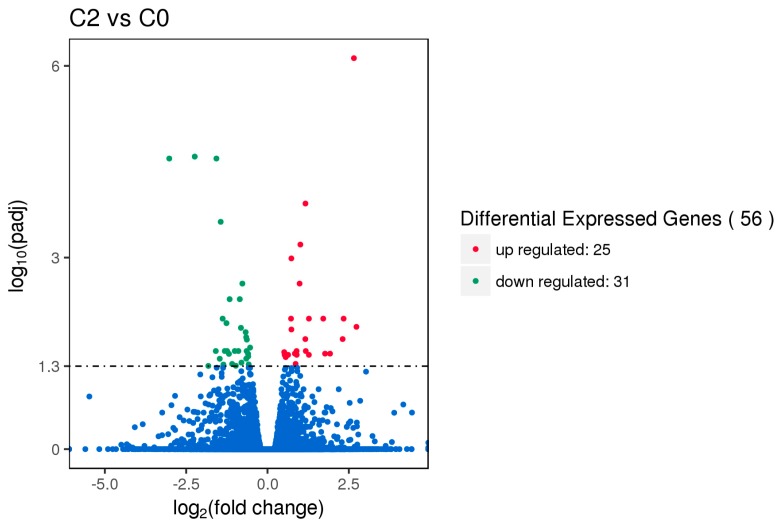
Volcano plot of differentially expressed genes in the pituitary. Differentially expressed genes (DEGs) were filtered using Padj < 0.05 as a threshold. Red spots represent up-regulated genes, and green spots indicate down-regulated genes. Blue spots represent genes that did not show obvious changes between high-egg production (HEP) and low-egg production (LEP) samples.

**Figure 3 animals-10-00090-f003:**
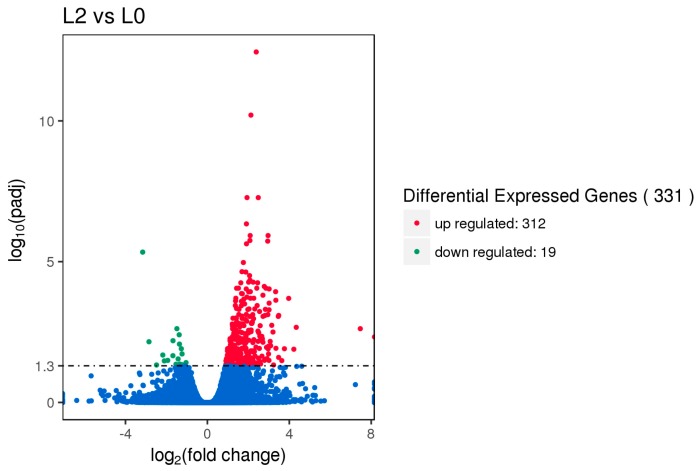
Volcano plot of differentially expressed genes in the ovary. DEGs were filtered using Padj < 0.05 as a threshold. Red spots represent up-regulated genes, and green spots indicate down-regulated genes. Blue spots represent genes that did not show obvious changes between the HEP and LEP samples.

**Figure 4 animals-10-00090-f004:**
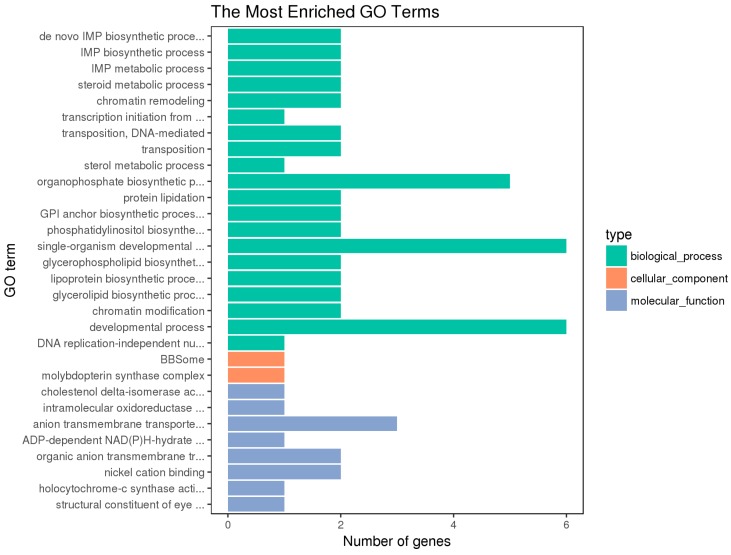
GO Enrichment of the Hypothalamic DEGs. The vertical coordinate is the enriched GO term, and the horizontal coordinate is the number of differential genes in the term. Different colors are used to distinguish biological processes, cell components, and molecular functions (the same below).

**Figure 5 animals-10-00090-f005:**
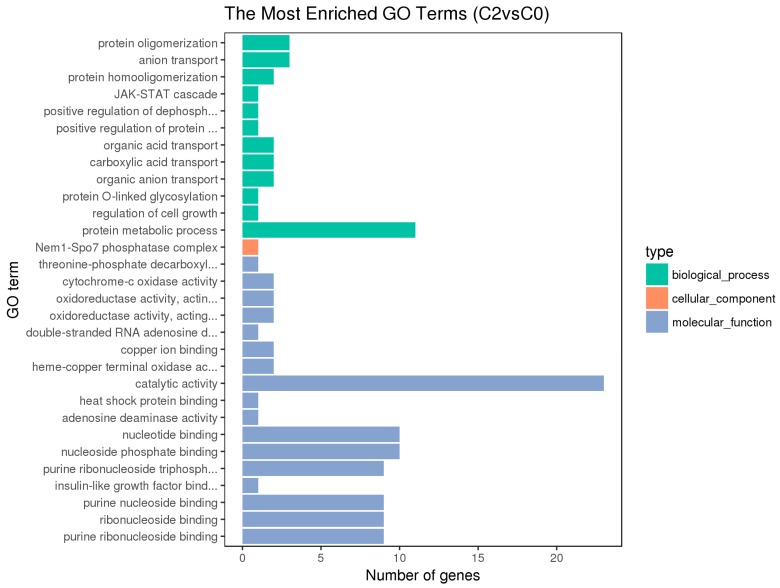
Histogram of GO Enrichment of the Pituitary DEGs.

**Figure 6 animals-10-00090-f006:**
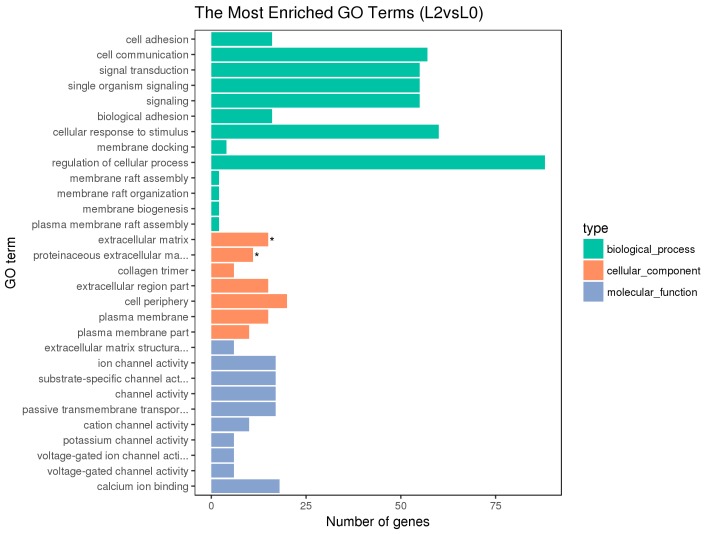
Histogram of GO enrichment of the ovarian DEGs. With “*” as the significantly enriched GO term.

**Figure 7 animals-10-00090-f007:**
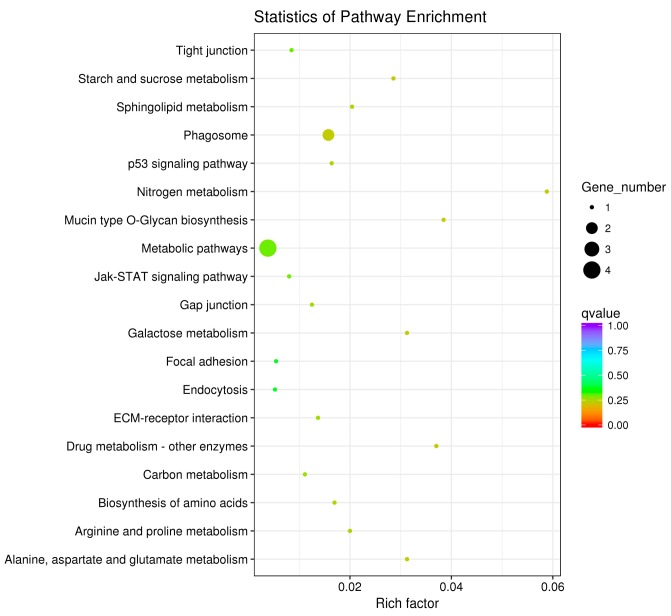
KEGG (Kyoto Encyclopedia of Genes and Genomes) Analyses of the hypothalamic DEGs. The vertical axis represents the pathway name, the horizontal axis represents rich factor, the size of the point represents the number of differentially expressed genes in this pathway, and the color of the point corresponds to different Q-value ranges (the same below).

**Figure 8 animals-10-00090-f008:**
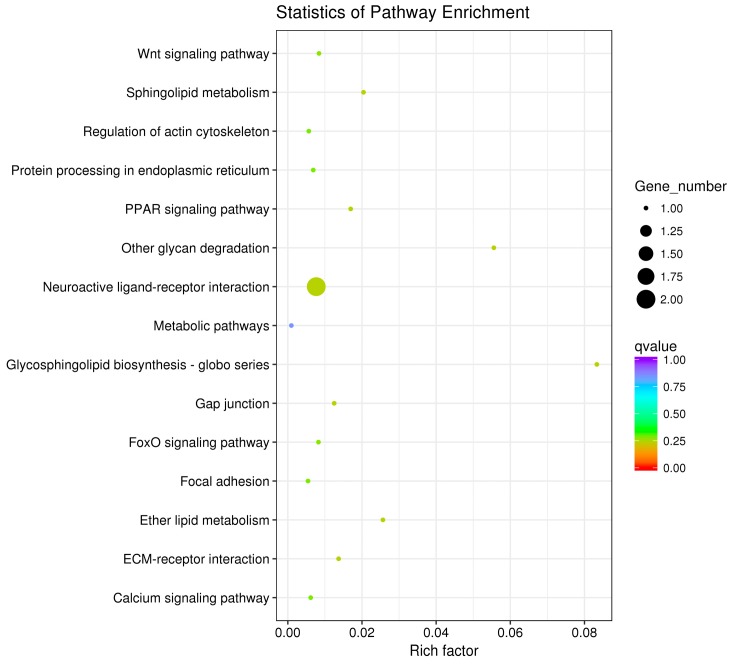
KEGG Analyses of the pituitary DEGs.

**Figure 9 animals-10-00090-f009:**
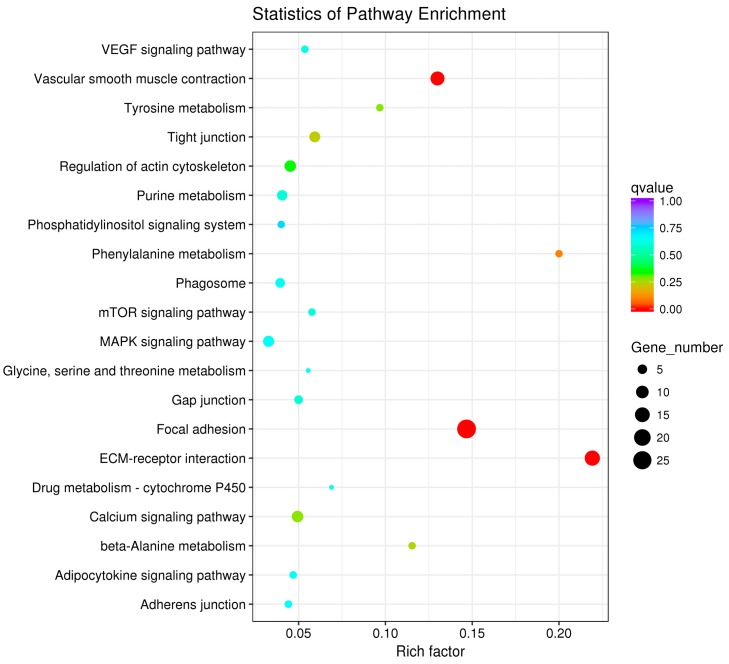
KEGG Analyses of the ovarian DEGs.

**Figure 10 animals-10-00090-f010:**
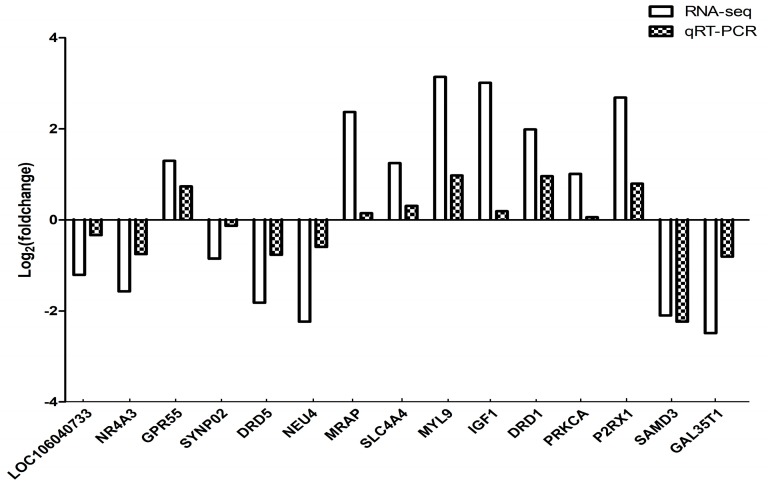
Comparison of qRT-PCR and RNA-seq Results. Total RNA extracted from the hypothalami, pituitary glands, and ovaries tissues that were measured by qRT-PCR analysis; relative expression levels were calculated according to the 2^−∆∆Ct^ method using beta-actin as an internal reference gene.

**Table 1 animals-10-00090-t001:** Alignment results of the clean reads to the reference genome.

Sample Name	X04	C04	L04	X23	C23	L23
Total reads	52,544,012	47,471,474	43,311,034	49,243,474	43,885,970	44,825,988
Total mapped	40,355,619 (76.8%)	36,619,486 (77.14%)	31,406,234 (72.51%)	38,401,158 (77.98%)	33,704,381 (76.8%)	33,641,532 (75.05%)
Multiple mapped	388,703 (0.74%)	349,245 (0.74%)	620,638 (1.43%)	374,387 (0.76%)	313,471 (0.71%)	640,850 (1.43%)
Uniquely mapped	39,966,916 (76.06%)	36,270,241 (76.4%)	30,785,596 (71.08%)	38,026,771 (77.22%)	33,390,910 (76.09%)	33,000,682 (73.62%)

Note: Alignment results of 6 samples to the reference genome (1) total reads: The statistics of the number of sequencing sequences filtered by sequencing data (clean data). (2) Total mapped: Statistics of the number of sequencing sequences that can be located on the genome; in general, if there is no pollution and the reference genome is selected properly, the percentage of this part of data is more than 70%. (3) Multiple mapped: Statistics of the number of sequencing sequences with multiple comparison positions on the reference sequence; the percentage of this part of data is generally less than 10%. (4) Uniformly mapped: Statistics of the number of sequencing sequences with unique alignment positions on the reference sequence.

**Table 2 animals-10-00090-t002:** Differentially expressed genes in the hypothalamus-pituitary-gonadal axis.

Organ		DEG
**Hypothalamus**	**Up-regulated**	*SFXN1, SGMS1, THAP5, SPRY1, COG2, LOC106031011, CARHSP1, Novel02312, LOC106031299, DMAP1, LOC106030886, MTERF4, SNRNP35, LOC106036439, Novel00604, Novel00813, LOC106048860, LOC106033299, PMP22, LOC106038373, CENPP, LOC106043368, TMEM255B, Novel02772, LOC106034340, LOC106032431, RBP1, STOML3, Novel02183, Novel00159, GRAP2, Novel00353, LOC106035108, CPS1, ANKRD1, Novel03069, IGFBP1, LOC106033972, LOC106035461, SPTBN5, FAM167B, Novel01797, GALNT8, LOC106037026, Novel00680, Novel00440, GPR55, Novel02518, LOC106044451, Novel02028, KIF18B, Novel02623, LOC106039706, ARMC12, LOC106046673, LOC106035746, SLC22A18, Novel00332, Novel01473, Novel02483, LOC106037068, LOC106049124, AGR2, Novel00634, Novel00776, Novel02517, LOC106030701, LOC106034203, LOC106032232, LOC106046797, LOC106038395, LOC106033669, LOC106039869, Novel00497, LOC106047183, Novel00809, LOC106047182, LOC106049953, LOC106046796.*
	**Down-regulated**	*LOC106043121, Novel00149, LOC106048725, NR4A3, EGR4, Novel00772, LOC106049845, SLA, LOC106033075, LOC106043342, 106032369, LOC106029709, Novel02265, Novel02348, LOC106036764, RUNX3, Novel01461, Novel00460, LOC106037894, Novel00595, BARD1, Novel02665, ANTXR1, Novel03067, OTOL1, LOC106032087, C1QTNF3, Novel01396, DDR2, MXRA5, LOC106049821, LOC106036261, LOC106039154, LOC106045306, FAM227B, LOC106040733, Novel01364, Novel02473, RFWD3, CGN, FILIP1L, CEMIP, LOC106032402, F13A1, Novel00314, SCRT2, LOC106033600, RNF213, LOC106032129, Novel02666, AQP9, Novel01931, LOC106042177, MAP7D3, NTRK3, TIE1.*
**Pituitary**	**Up-regulated**	*LOC106031011, CRACR2B, SOX9, LOC106036438, LOC106036439, LOC106036974, TNC, NSMF, LOC106037323, ATP13A3, LOC106039850, DNAJA1, PRKG2, SPARCL1, P2RY2, LOC106042825, SLC4A4, TMEM201, SFRP4, LOC106044451, LOC106049066, Novel00497, Novel01686, Novel02339, Novel02920.*
	**Down-regulated**	*VIT, SLC15A2, LOC106031282, LOC106031667, FABP7, DRD5, NDE1, LOC106034326, AFAP1L1, LOC106034774, KCTD4, ENPP6, CTHRC1, LOC106036254, LOC106036743, SCIN, LOC106037776, SYNPO2, LOC106038998, NEU4, LOC106039845, LOC106042210, STK17A, MRAP, DNAJC28, SGK2, SPTSSB, SLAIN1, GBGT1, LOC106047046, LOC106049843.*
**Ovary**	**Up-regulated**	*LOC106029456, LOC106029515, LOC106029546, LOC106029652, LOC106029721, LOC106029781, SPEG, MYLK, HEG1, LOC106029988, SORBS1, LOC106030002, HOXA3, HOXA5, PRKG1, NR1D2, LRRC3B, LOC106030346, LOC106030347, CD109, KCNQ5, RIMS1, PSD, BAG2, LOC106030416, GEM, LOC106030538, PI15, COLEC11, GABRA2, RERG, LOC106030915, MICAL3, KIAA1671, PROX1, CDC42BPA, ACTN1, MEIS2, CACNA1H, MPRIP, LOC106032203, RASD1, KIAA0922, NPY2R, LOC106032455, LAMA2, FAM26E, LAMA4, ZCCHC24, PPP2R2B, KCNMA1, RASGEF1C, ADAMTS2, ACOT11, CDKN2C, ROR1, MAST2, ZSWIM5, VCL, PRKAA2, LOC106033331, RGS11, MYH11, CPN1, EEF2K, PRKAR1B, FBXL22, HIPK3, SYDE2, ADGRL4, KCNMB1, LOC106034392, LOC106034406, COL6A3, COL3A1, PRKCA, AFAP1, EMILIN2, LOC106035048, DGKH, SLC25A4, NCALD, SYBU, LOC106035506, SEMA3C, MAST4, PDE4D, RAB3C, INPP4A, RBFOX2, ALDH1L2, IGF1, PVALB, LOC106035922, GALR1, EML1, KCNE4, ADAMTS12, TESK1, RAI14, TPM2, RXFP3, LOC106036401, SYNM, NR2F2, LOC106036475, RGMA, ALDH1A3, CILP, RASL12, LOC106036832, LOC106036842, ZBTB47, LOC106037166, AKAP5, ADSSL1, RHOJ, DACT1, SETBP1, POPDC2, ARL13A, IL13RA1, PON2, GREB1L, LOC106037870, SYNC, LOC106037900, LDLRAP1, MGLL, ADCY3, ABCC9, LOC106038240, KCNJ8, LIMD1, FYCO1, ZHX1, LOC106038471, LOC106038505, IRS4, CHRDL1, ABI3BP, COL4A6, COL4A5, CAPN6, SYNPO2, TGFBR3, CDK6, CLMP, GAB2, LOC106039897, SDPR, LOC106040026, LOC106040036, SLC6A1, FBLN2, LACTBL1, LIMS2, GPR17, MYO7B, LOC106040291, CRYL1, LOC106040422, ADA, JPH2, JAM3, ADAMTS8, MORN5, LOC106040733, SCUBE2, NRIP3, MICAL2, MRVI1, HSPB1, LOC106041406, MYO1C, P2RX1, FMO3, PRRX1, RGS5, LOC106041605, FHL2, CAP2, SUSD5, LAMB1, CAV1, LOC106042166, GPR20, CCDC102B, SLC24A3, LOC106042361, ARHGEF17, CHRDL2, RGS6, ANGPTL1, ZBTB37, ZEB1, KIAA1462, LOC106042668, PDLIM7, NEURL1B, DRD1, LOC106042807, MUSTN1, STARD9, INHBA, PGM5, SPTBN5, GNAO1, LOC106043213, LOC106043342, CEMIP, MRAP, LOC106043529, PDE4B, RAVER2, TEK, PITPNM3, MN1, SLC4A4, FERMT2, LOC106044084, ADAMTSL2, CSPG4, PEAK1, LINGO1, LOC106044229, KIF1B, ACTA2, ADGRA2, COL4A1, COL4A2, LOC106044472, LOC106044704, MEIS1, MUC4, MASP1, CDH13, LOC106045192, OTOGL, MYL9, LOC106045455, MED13L, LOC106045561, LOC106045776, LOC106045852, LOC106045870, CSRP1, TNS1, MYL4, LOC106046005, IP6K3, TEAD3, LMOD1, PRDM6, LOC106046391, PPP1R12A, 106046568, LOC106046709, TRHDE, LOC106047053, MSRB3, MYOCD, NCS1, PRR16, SMTN, CCDC50, UTS2B, ACTG2, SCNN1A, PIANP, LOC106048001, LOC106048002, PRSS23, MFAP5, CPT1B, HOXB3, THBS3, DNAJB5, IL11RA, HSPG2, KCNK1, ACTA1, NID1, LOC106048884, FLNC, NPR1, ECM2, OGN, MYO18A, COL6A1, PCBP3, COL6A2, GLI1, Novel00206, Novel00225, Novel00283, Novel00349, Novel00428, Novel00498, Novel00508, Novel00568, Novel00676, Novel00688, Novel00911, Novel01277, Novel01584, Novel01601, Novel02245, Novel02466, Novel02966, Novel03036, Novel03063.*
	**Down-regulated**	*KCNB2, SAMD3, FSTL4, LOC106033036, PRRG4, CA10, LOC106035557, LOC106036685, LOC106042047, LOC106042103, TSPAN8, LOC106046958, LOC106048733, LOC106048993, GAL3ST1, Novel00233, Novel02085, Novel02711, Novel02751.*

**Table 3 animals-10-00090-t003:** Putative candidate genes of the DEGs associated with geese laying.

Terms/Pathways	Gene Name	log2FoldChange	Padj	Up-Down Regulation (HEP/LEP)
steroid biosynthetic process	*LOC106035461*	1.247	0.73322	Up
steroid hormone-mediated signaling pathway	*NR4A3*	−1.5699	0.34746	Down
developmental process	*LOC106040733*	−1.2066	0.71683	Down
reproduction	*Novel01396*	−1.2601	0.72516	Down
G-protein coupled receptor signaling pathway	*GPR55*	1.3013	0.7477	Up
growth	*LOC106034326*	−1.0064	0.028997	Down
G-protein coupled receptor signaling pathway	*P2RY2*	1.7694	0.031867	Up
	*DRD5*	−1.8187	0.049623	Down
calcium ion binding	*CRACR2B*	1.7139	0.0090239	Up
	*SPARCL1*	2.7338	0.012128	Up
reproduction	*HOXB3*	1.1354	0.028618	Up
	*DRD1*	1.9867	0.028259	Up
developmental process	*LAMA2*	1.8842	0.0060556	Up
	*LAMA4*	1.4628	0.0014504	Up
	*COL4A1*	2.388	0.039715	Up
	*COL4A2*	2.2348	0.033559	Up
	*COL4A5*	1.0104	0.018881	Up
	*P2RX1*	2.6862	0.0054992	Up
	*ADCY3*	1.6102	0.0060191	Up
	*LOC106040733*	1.3839	0.026521	Up
Neuroactive ligand-receptor interaction	*DRD5*	−1.8187	0.049623	Down
	*P2RY2*	1.7694	0.031867	Up
Sphingolipid metabolism	*NEU4*	−2.237	0.0000000023607	Down
Focal adhesion	*IGF1*	3.0141	0.00029768	Up
Oocyte meiosis				
Progesterone-mediated oocyte maturation				
mTOR signaling pathway				
FoxO signaling pathway				
p53 signaling pathway				
Vascular smooth muscle contraction	*PRKCA*	1.0063	0.023648	Up
Calcium signaling pathway				
MAPK signaling pathway				
Tight junction				
Gap junction				
mTOR signaling pathway				
VEGF signaling pathway				
Phosphatidylinositol signaling system				
Focal adhesion				
Melanogenesis				
Vascular smooth muscle contraction	*ADCY3*	1.6102	0.0060191	Up
Calcium signaling pathway				
GnRH signaling pathway				
Oocyte meiosis				
Progesterone-mediated oocyte maturation				
Tight junction				
Gap junction				
Purine metabolism				
Adrenergic signaling in cardiomyocytes				
Melanogenesis				
Gap junction	*PRKG1*	1.3621	0.0018268	Up
Focal adhesion	*THBS3*	2.4123	0.000087642	Up
ECM-receptor interaction				
Phagosome				
Vascular smooth muscle contraction	*MYLK*	2.8068	0.00081465	Up
Calcium signaling pathway				
Tight junction				
Focal adhesion				
Regulation of actin cytoskeleton				
Focal adhesion	*MYL9*	3.1427	0.0018211	Up
Vascular smooth muscle contraction				
Regulation of actin cytoskeleton				
Calcium signaling pathway	*LOC106045455*	2.4292	0.00036177	Up
Tight junction				
Tight junction	*SLC25A4*	2.7157	0.00087784	Up
Calcium signaling pathway				
Calcium signaling pathway	*CACNA1H*	1.4128	0.0014504	Up
MAPK signaling pathway				
Tight junction				

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
