# Peer review of "Transcriptomic Analyses of the Hypothalamic-Pituitary-Gonadal Axis Identify Candidate Genes Related to Egg Production in Xinjiang Yili Geese"

_animals, 2020, doi:10.3390/ani10010090_

Round 1

Reviewer 1 Report

In order to improve the egg production performance of Xinjiang Yili geese, the authors used high-throughput transcriptome sequencing technology to compare and analyze differentially expressed genes relevant to the hypothalamic-pituitary-gonadal axis of Xinjiang Yili geese with high or low egg production, and identified candidate genes that are related to egg production of Xinjiang Yili geese. 135, 56, and 331 differentially expressed genes were identified in the hypothalamus, pituitary, and ovary, respectively. Of which, 15 genes were randomly selected and confirmed by using qRT-PCR method. This study, for the first time, provided the transcriptomic information of the hypothalamic-pituitary-gonadal axis of Xinjiang Yili geese and laid the theoretical basis for revealing the molecular mechanisms regulating the egg-laying traits of Xinjiang Yili geese. Generally this study is well designed, and the writing of this manuscript is clearly organized. However, before being accepted for publication, the authors need to address the issues as the following.

Line 79-91, in “Sample Collection” section, (1) ages, body weights of the geese sampled should be provided. (2) goose raising conditions and method should be given.

Line 136, in the “……, 58 ° C for 20 s,……” , the annealing temperature, 58 ° C is suitable for the 15 candidate genes?

Line 129, “Fourteen differentially expressed genes related……,” should be modified as “Fifteen differentially expressed genes related……”

Line 141, 33 of primers were list in Table 1, please check it. WHY?

GenBank accession number of the NCBI Reference Sequences for all the candidates (includingβ-actin gene) should be supplemented in the Table.

Line 227, Line 282, “RT-PCR……,” should be changed as “qRT-PCR……”

All images used in the Figures (from figure1 to figure 10) should be replaced by clear one.

Author Response

Response to Reviewer 1 Comments

Point 1: I found that Line 79-91, in “Sample Collection” section, (1) ages, body weights of the geese sampled should be provided. (2) goose raising conditions and method should be given.

Response 1: (1)  The age and weight of the experimental Yili geese have been supplemented.

The feeding method of Yili Goose has been added

Point 2: Line 136, in the “……, 58 ° C for 20 s,……” , the annealing temperature, 58 ° C is suitable for the 15 candidate genes?

Response 2: The annealing temperature of the 15 differentially expressed genes is in the range of 58~60 ℃, and the annealing temperature of the differentially expressed genes has been described in detail.

Point 3:Line 129, “Fourteen differentially expressed genes related……,” should be modified as “Fifteen differentially expressed genes related……”

Response 3:“Fifteen differentially expressed genes related……” have been modified

Point 4: Line 141, 33 of primers were list in Table 1, please check it. WHY?

GenBank accession number of the NCBI Reference Sequences for all the candidates (includingβ-actin gene) should be supplemented in the Table.

Response 4: The primer information has been checked and modified, and the GenBank accession number of the NCBI Reference Sequences for all the candidates has been added

Point 5: Line 227, Line 282, “RT-PCR……,” should be changed as “qRT-PCR……”

Response 5: Relevant information has been modified

Point 6: All images used in the Figures (from figure1 to figure 10) should be replaced by clear one.

Response 6: Replaced with clear picture

Reviewer 2 Report

I have reviewed the manuscript entitled “Transcriptomic Analyses of the Hypothalamic-Pituitary-Gonadal Axis Identify Candidate Genes Related to Egg Production in Xinjiang Yili Geese” for a publication in Animals as an original article. The authors investigated RNA-seq from three different tissues, which are hypothalamus, pituitary, and ovary from high-egg production (HEP, n = 4) and low-egg production (LEP, n = 4) and real-time PCR validation. Analyses and results look nice. However, I think there are some discrepancies in their description. In addition, almost all Figures look low-resolution and letter size should be larger. Therefore, I recommend this paper needs to be revised before publishing. My minor comments will be shown below:

Minor comments

1: In Line 16, 19, 22, there are three-types of breed name “Xinjiang Yili Geese”, “Xinjiang Ili Geese” and “Xinjiang Yil Geese”. These all should be “Xinjiang Yili Geese”.

2: In Line 129, “toHEP” should be “to HEP”.

3: In Line 134, µmol L/L is correct? I think µmol/L will be correct.

4: In Line 145, the authors described “integrity of 8 RNA samples”. However, only 6 samples were shown in Table 2. What is the definition of the “sample”?

5: In Line 149, “8 samples was shown in Table 3” was described. But, In Table 3, there are 24 rows. What is the definition of the “sample”?

6: In Line 168, the authors also described “eight samples … in Table 4”. But, in Table 4, there are 6 columns and footnote indicates “8 samples”. What is the discrepancy?

7: In Line 188, the authors used p-value < 0.005 only for the hypothalamus. Since the authors described the threshold FDR corrected p-value (Padj < 0.05) in Materials and Methods, Padj threshold should be used. If not, I would like to know why p-value < 0.005 was used only for the hypothalamus. Anyway, the authors should describe that p-value < 0.005 was used only for the hypothalamus in Materials and Methods.

8: In Line 194, “Figure 1, Figure 2, Figure 3” should be “Figures 1-3”.

9: In Figure 1, the letter size of title of the bar and legend were too small. Please set the same letter size in Figures 2 and 3.

10: In Line 224, “nucleoside” would be “nucleotide”.

11: In Figures 4-9, the letter size of legend should be larger. This version of the Figures is inadequate.

12: In Figure 10, fold-change was shown. However, I did not recognize which one (HEP?) is the standard. What means if fold change is positive? Figure legend should be used for the explanation.

13: All of the Figures should be added more explanations. The present version contains only the title of the Figure. Legend should be added for the potential readers.

14: In Line 364, “secretion.By” should be “secretion. By”.

Author Response

Response to Reviewer 2 Comments

Point 1: In Line 16, 19, 22, there are three-types of breed name “Xinjiang Yili Geese”, “Xinjiang Ili Geese” and “Xinjiang Yil Geese”. These all should be “Xinjiang Yili Geese”.

Response 1: The name of experimental animal has been modified as “Xinjiang Yili Geese”.

Point 2: In Line 129, “toHEP” should be “to HEP”.

Response 2: The inappropriate format has been modified

Point 3: In Line 134, µmol L/L is correct? I think µmol/L will be correct.

Response 3:I'm sorry for my negligence, modified to the correct international unit

Point 4: In Line 145, the authors described “integrity of 8 RNA samples”. However, only 6 samples were shown in Table 2. What is the definition of the “sample”?

Response 4: I'm sorry for my mistake, I want to describe RNA quality testing of 24 samples from 8 individuals(Samples of hypothalamus, hypophysis and ovary of 8 Xinjiang Yili geese with high and low egg production), and only 6 of them are shown in the table.

Point 5: In Line 149, “8 samples was shown in Table 3” was described. But, In Table 3, there are 24 rows. What is the definition of the “sample”?

Response 5: The quality assessment of sequencing output data modified to 24 samples was shown in table 3

Point 6:  In Line 168, the authors also described “eight samples … in Table 4”. But, in Table 4, there are 6 columns and footnote indicates “8 samples”. What is the discrepancy?

Response 6: modified to “The statistics of the sequencing read alignments of the 24 samples to the reference genome are shown in Table 4.” The table only shows the comparison of 6 samples with the reference genome.

Point 7: In Line 188, the authors used p-value < 0.005 only for the hypothalamus. Since the authors described the threshold FDR corrected p-value (Padj < 0.05) in Materials and Methods, Padj threshold should be used. If not, I would like to know why p-value < 0.005 was used only for the hypothalamus. Anyway, the authors should describe that p-value < 0.005 was used only for the hypothalamus in Materials and Methods.

Response 7: Only one candidate gene was screened when Padj < 0.05 gene was selected as differential expression gene in the hypothalamus.Therefore, the adjusted threshold for differentially expressed genes in the hypothalamus was p-value <0.005.

Relevant content has been supplemented in the material method.

Point 8: In Line 194, “Figure 1, Figure 2, Figure 3” should be “Figures 1-3”.

Response 8: “Figures 1-3”have been modified

Point 9:  In Figure 1, the letter size of title of the bar and legend were too small. Please set the same letter size in Figures 2 and 3.

Response 9: The format of the diagram has been modified

Point 10: In Line 224, “nucleoside” would be “nucleotide”.

Response 10: “nucleotide”have been modified

Point 11: In Figures 4-9, the letter size of legend should be larger. This version of the Figures is inadequate.

Response 11: The legend has been modified

Point 12: In Figure 10, fold-change was shown. However, I did not recognize which one (HEP?) is the standard. What means if fold change is positive? Figure legend should be used for the explanation.

Response 12: A positive fold change indicates that the differentially expressed gene is an up-regulated gene, otherwise it is a down-regulated gene. The RNA-seq trend is consistent with the trend of fluorescence quantitative results, indicating that the RNA-seq results are reliable.

Point 13: All of the Figures should be added more explanations. The present version contains only the title of the Figure. Legend should be added for the potential readers.

Response 13: Legend has been added to all pictures

Point 14: In Line 364, “secretion.By” should be “secretion. By”.

Response 14: The format has been modified

Reviewer 3 Report

The manuscript by Wu et al. reported the gene expression differences in hypothalamus, pituitary and ovary tissues of Xinjiang Yil geese with egg performance using RNA-seq. The Go and KEGG functional pathway analysis found that DEGs were involve in gap junction, focal adhesion, and ECM-receptor interaction signaling pathways hypothalamic, pituitary, and ovarian axis. There are, however, several aspects of the manuscript that are immature overall which is given as follows:

As this article is mainly focused on high and low production of egg in Yili Geese, I am wondering why authors did not studied several egg production traits during the laying period. Authors claim that they identified 30 candidate genes related to egg production, however I am unable to see these genes in the manuscript. Add scientific name of "Yili geese" when presenting first time in the article also can be given in the title. Page 1, line33: remove space in (Nr ) Typo: Name of the animal studied is very confusing, sometimes it is Yili Geese, Yili goose, ili Geese or Yil Geese such as in page1 line 16 and 19, it should be Yili Geese in entire manuscript. Give a precise introduction including egg production related study in other birds. All gene and protein names need to be checked carefully, as gene names are italic but protein names should be written in plain front. Ethical statement needed for this study. “The total RNA was extracted using Trizol by following the manufacturer’s protocol.”, provide the manufacturer name. A suitable reference should be cited for CASAVA, HISAT, HTSeq software and provide the necessary reference for other software used in methods. Page3 line113: replace “must be” to “were”. line 129: “toHEP”, add space. line137, remove space. line138: change 2'-ΔΔct> 2-ΔΔct. The methodology used to screen DEGs is not appropriate which also reduced the weight of this paper, I suggest authors to explain in details. It is not mentioned in the manuscript, how many replicates were used to perform quantitative real-time PCR? Table 1: NCBI serial number should be NCBI accession number or GenBank accession number Page7 line188: Why authors used “Padj < 0.05” for pituitary and ovary while “p-value < 0.005” for hypothalamus. Figures (1-10) are not clear with good quality Table 2: it is difficult to understand the table as it is not arranged well. Table 1-3 can be given as supplementary information. Figure 10: provide the standard error bar on the graph of each gene. Authors majorly discussed the genes related to G protein-coupled receptors, while it is important to focus on some important genes related to egg production that identified in this study.

Author Response

Response to Reviewer 3 Comments

Point 1: As this article is mainly focused on high and low production of egg in Yili Geese, I am wondering why authors did not studied several egg production traits during the laying period.

Response 1: This research was carried out on the basis of the nutritional needs of the Yili goose in the early stage of the research group. In the early stage of the research team, the egg production, egg production rate, egg quality and other aspects of the Yili goose have been studied.(http://kns.cnki.net//KXReader/Detail?TIMESTAMP=637122704003365000&DBCODE=CJFQ&TABLEName=CJFDLAST2019&FileName=DWYX201904020&RESULT=1&SIGN=9SaPd1lvCpM5KdT6vO1S38wVGMk%3d)

Point 2: Authors claim that they identified 30 candidate genes related to egg production, however I am unable to see these genes in the manuscript.

Response 2: I will provide the information about the 30 candidate genes

Point 3: Add scientific name of "Yili geese" when presenting first time in the article also can be given in the title. Page 1, line33: remove space in (Nr ) Typo: Name of the animal studied is very confusing, sometimes it is Yili Geese, Yili goose, ili Geese or Yil Geese such as in page1 line 16 and 19, it should be Yili Geese in entire manuscript.

Response 3: The inappropriate format has been changed and the name of the test animal has been changed to Yili geese

Point 4: All gene and protein names need to be checked carefully, as gene names are italic but protein names should be written in plain front.

Response 4: The gene name and protein name have been checked

Point 5: Ethical statement needed for this study.

Response 5: Ethical statement has been added. All of the animals mentioned in this study were approved by the Ethics of Xinjiang Agriculture University(Approval number :2017011).

Point 6: “The total RNA was extracted using Trizol by following the manufacturer’s protocol.”, provide the manufacturer name.

Response 6: The manufacturer name is Invitrogen,USA

Point 7: A suitable reference should be cited for CASAVA, HISAT, HTSeq software and provide the necessary reference for other software used in methods.

Response 7: Reference is provided to the software used in the method

Point 8: Page3 line113: replace “must be” to “were”. line 129: “toHEP”, add space. line137, remove space. line138: change 2'-ΔΔct> 2-ΔΔct. The methodology used to screen DEGs is not appropriate which also reduced the weight of this paper, I suggest authors to explain in details.

Response 8: The inappropriate format has been corrected.

Point 9: It is not mentioned in the manuscript, how many replicates were used to perform quantitative real-time PCR?

Response 9: Fifteen differentially expressed genes related to HEP and LEP of Xinjiang Yili geese were randomly selected from the transcriptome sequencing results for fluorescence-based quantitative validation.

Point 10: Table 1: NCBI serial number should be NCBI accession number or GenBank accession number

Response 10: NCBI accession number has been added

Point 11: Page7 line188: Why authors used “Padj < 0.05” for pituitary and ovary while “p-value < 0.005” for hypothalamus.

Response 11: Only one candidate gene was screened when Padj < 0.05 gene was selected as differential expression gene in the hypothalamus.Therefore, the adjusted threshold for differentially expressed genes in the hypothalamus was p-value <0.005.

Relevant content has been supplemented in the material method.

Point 12: Figures (1-10) are not clear with good quality

Response 12: Replaced with clear Figures(1-10)

Point 13: Table 2: it is difficult to understand the table as it is not arranged well.

Response 13: I'm sorry for my mistake, I want to describe RNA quality testing of 24 samples from 8 individuals(Samples of hypothalamus, hypophysis and ovary of 8 Xinjiang Yili geese with high and low egg production), and only 6 of them are shown in the table.

Point 14: Table 1-3 can be given as supplementary information.

Response 14: Tables 1-3 have been adjusted to Tables S1-S3

Point 15: Authors majorly discussed the genes related to G protein-coupled receptors, while it is important to focus on some important genes related to egg production that identified in this study.

Response 15: The genes mainly discussed were those enriched with multiple GO terms or involved in multiple pathways, which were speculated to be related to the egg-laying traits of yili geese.

Round 2

Reviewer 2 Report

The authors corrected well. 

Author Response

Thank you for your comments and suggestions, which is very helpful to me.

Reviewer 3 Report

The authors addressed well my all points raised in the original manuscript and greatly improved the manuscript. However, the paper still suffers from many clerical errors, thus, I wrote down some corrections must be addressed.

Comments to authors:

1- Line 80: “According to the continuous and complete egg production records”, complete egg production is not accurate, give a complete laying period of Yili goose in days at which authors recorded total egg production.

2- Line 80: Provide the method of ovarian tissue sample collection, as follicles consists of egg yolk/fat tissues which might reduce transcriptome data quality.

3- In response to my previous comment 2 as authors claim to identify 30 candidate genes, they replied “I will provide the information about the 30 candidate genes”, I am wondering as it is key finding of this study but I am unable to find this data in the manuscript, which must be addressed.

4- Line 132: delete “.” Along with this many typing errors are found in manuscript, make sure it should be corrected.

5- Line 152: replace “canbe found” with “are given”.

6- Line 190: Figure 1- 3 should be Figures 1- 3

7- Table 2: should be restructured in view of easy to understand for readers and also check “、-//-”.

8- In figure 4 legend: authors mention that "*"indicating significant enriched GO term, however I couldn’t see "*" in figure 4, however it is only shown in figure 6. It should be corrected.

9- Line 251: delete “.” from Figures. 7-9

10- Line 289: change 2'-ΔΔCt to 2-ΔΔCt

11- Line 289: “Slc4a4”, follow the same pattern in entire manuscript.

Author Response

Response to Reviewer 3 Comments

Point 1: Line 80: “According to the continuous and complete egg production records”, complete egg production is not accurate, give a complete laying period of Yili goose in days at which authors recorded total egg production.

Response 1: Yili geese lays eggs seasonally, with only one laying period every year. Generally, egg breeding occurs in February to June, and the peak egg production period is from March to April.(Related content has been added)

Point 2: Line 80: Provide the method of ovarian tissue sample collection, as follicles consists of egg yolk/fat tissues which might reduce transcriptome data quality.

Response 2: In order to ensure the consistency of the samples collected.Quickly collect the ovary samples of Xinjiang Yili goose after slaughter, which comprised the whole ovary including the small and large yellow follicles. and then after rinsing 1-2 times with PBS, immerse the sample completely in RNA later, and immediately aliquot and store it in RNA later Label the solution in a cryopreserved tube, leave it at 4 ℃ overnight (so that the protective solution penetrates into the tissue), and then store it at -80℃ for the extraction of total RNA from tissue samples.

Point 3: In response to my previous comment 2 as authors claim to identify 30 candidate genes, they replied “I will provide the information about the 30 candidate genes”, I am wondering as it is key finding of this study but I am unable to find this data in the manuscript, which must be addressed.

Response 3: Table 3 shows the information of 30 candidate genes speculated in this studyTable 3 Putative candidate genes of the DEGs associated with geese laying”

Point 4: Line 132: delete “.” Along with this many typing errors are found in manuscript, make sure it should be corrected.

Response 4: The inappropriate format has been corrected.

Point 5: Line 152: replace “canbe found” with “are given”.

Response 5: Incorrect grammar has been corrected.

Point 6: Line 190: Figure 1- 3 should be Figures 1- 3.

Response 6: The inappropriate format has been corrected.

Point 7: Table 2: should be restructured in view of easy to understand for readers and also check “、-//-”.

Response 7: Relevant contents have been checked and modified.

Point 8: In figure 4 legend: authors mention that "*"indicating significant enriched GO term, however I couldn’t see "*" in figure 4, however it is only shown in figure 6. It should be corrected.

Response 8: “*”have been shown in figure 6.

Point 9: Line 251: delete “.” from Figures. 7-9.

Response 9: The inappropriate format has been corrected.

Point 10: Line 289: change 2'-ΔΔCt to 2-ΔΔCt.

Response 10: The inappropriate format has been corrected.

Point 11: Line 289: “Slc4a4”, follow the same pattern in entire manuscript.

Response 11: The same pattern has been unified for SLC4A4.
